**Data Availability Statement:** The authors confirm that, for approved reasons, some access restrictions apply to the data underlying the

# Experiences of a web-based psycho-educational intervention targeting sexual dysfunction and fertility distress in young adults with cancer—A self-determination theory perspective

**Claire Micaux Obol**[1]*, **Claudia Lampic**[1,2], **Lena Wettergren**[1], **Lisa Ljungman**[1,3], **Lars E. Eriksson**[4,5,6]

1 Department of Women's and Children's Health, Karolinska Institutet, Stockholm, Sweden, 2 Department of Public Health and Caring Sciences, Uppsala University, Uppsala, Sweden, 3 Department of Women's and Children's Health, Uppsala University, Uppsala, Sweden, 4 Department of Learning, Informatics, Management and Ethics, Karolinska Institutet, Stockholm, Sweden, 5 School of Health Sciences, City, University of London, London, United Kingdom, 6 Department of Infectious Diseases, Karolinska University Hospital, Huddinge, Sweden

* claire.micaux.obol@ki.se

## Abstract

### Introduction

Sexual and reproductive health are significant aspects of quality of life. Healthcare often fails to provide adequate support for young cancer survivors in this area, hence the need to develop more effective interventions. The present study aimed to describe experiences of participating in a web-based psycho-educational intervention focusing on sexual dysfunction and fertility distress after cancer, and to explore these experiences within the theoretical frame of the basic psychological needs for competence, relatedness and autonomy according to self-determination theory.

### Methods

Individual semi-structured interviews with 24 women and 4 men, age 19–40, were abductively analyzed using the Framework approach for qualitative content analysis.

### Results

Participant experiences corresponded well with the three main deductive themes competence, relatedness and autonomy, divided into a total of nine subthemes illustrating varying degrees of basic need satisfaction with considerable nuance but not without ambiguity. While satisfaction of the need for competence could be linked to the amount of information in relation to participants' cognitive capacity, satisfaction of the need for relatedness seemed to be of special importance for these young adults with cancer experience. Invitation to the program meant a chance at alleviating loneliness and normalizing problems, symptoms and

findings. Due to ethical (permit no 2013/1746-31/4, Regional Board of Ethics, Stockholm) and legal (GDPR regulation) constraints, we are not allowed to deposit or publicly share the raw data. We do not have the participants' consent to share data that are potentially identifying and that may contain private and/or sensitive information. Data are available on request from: Michael Rosendahl (not an author of the present manuscript), Head of archives, Department Women's and Children's Health, Karolinska Institutet, SE-17177 Stockholm, Sweden, for researchers who meet the criteria for access to confidential data.

**Funding:** CMO: The Swedish Cancer Society (CAN 2016/615) https://www.cancerfonden.se/ The Doctoral School in Health Care Sciences at Karolinska Institutet (3-780/2014) https://ki.se/nvs/forskarskolan-i-vardvetenskap CL: The Swedish Research Council for Health, Working Life and Welfare (2014-4689) https://forte.se/ LW: The Swedish Childhood Cancer Foundation (TJ2014-0050) https://www.barncancerfonden.se/ L.L.: The Swedish Research Council (2017-01530) https://www.vr.se/ L.E.E.: No specific funding for this work The funders had no role in study design, data collection and analysis, decision to publish, or preparation of the manuscript.

**Competing interests:** The authors have declared that no competing interests exist.

concerns. Participants' descriptions of perceived autonomy support were more challenging and ambiguous, because of the many contradictions in participants' responses to their variable situations.

## Conclusion

Basic psychological needs were confirmed as flexible positions along a continuum rather than discrete and mutually exclusive qualities. Understanding the variety of basic need satisfaction may enhance the design of future web-based interventions to be even more inclusive, tailorable and autonomy-supportive. Further research is warranted to determine the role of basic need satisfaction as a possible mediator for web-based psychoeducational interventions in cancer survivorship care.

## Introduction

Sexual and reproductive health deeply affects people's wellbeing and quality of life and needs to be further integrated into the psychosocial care of young adults (age 19–40) with cancer. About 50% of young adult survivors report negative impact on sexual function approximately one year after cancer diagnosis [1]. Chemotherapy, radiation and hormonal therapy all affect the body's mucous membranes and may cause vaginal dryness in women. Circulatory changes due to surgery or radiation in the genital area may lead to neural damage, which is why some men experience erectile dysfunction and both women and men may suffer from diminished sensitivity, hence orgasm difficulties, after cancer treatment [2]. These physiological changes combined with psychosocial factors such as alterations of body image and long-lasting fatigue [1], can contribute to lower satisfaction with sexual function (women) and lack of desire (men) through young adulthood [3] and even in the long term [4]. In addition, several cancer treatments affecting reproductive organs or the hormonal system may cause temporary or permanent subfertility in both women and men [5]. Experiencing relationship and intimacy difficulties and/or the threat of infertility may be deeply distressing [6] and has been associated with mental health issues [7]. Research suggests that healthcare fails to provide adequate support and treatment for these and other survivorship concerns [8, 9].

Internet interventions have the potential to provide flexible, cost-efficient support [10] but, given the variation in trial designs, their overall effectiveness remains contested [11–13]. To be effective, interventions need to have an explicit theoretical orientation relevant for the intended outcome [14].

According to the self-determination theory by Ryan and Deci [15] the three universal basic psychological needs for competence, relatedness and autonomy have to be nurtured in order to promote motivation and optimal human functioning. Differences in the degree of satisfaction of these basic psychological needs will largely predict differences in well-being. In order to understand ill-being, it is of essence to also study factors *thwarting* those basic needs [16]. The need for *competence* involves adaptive learning of skills that enable the individual to control the outcome of actions, i.e. feeling able. *Relatedness* concerns the social context and the individual's striving towards a reciprocal feeling of belonging with others. The concept of *autonomy* refers to volitional, self-endorsed and authentic, as opposed to controlled, actions. Supporting the autonomy of patients does not mean leaving them alone with difficult decisions, but rather providing them with tools to make informed, mindful and reflective choices,

respecting these choices and engaging with the ensuing challenges [15, 17]. In healthcare interventions, autonomy support has shown to be a predictor for satisfaction of competence and relatedness needs [17] and to be a potential mediator explaining both behavioral and health outcomes [18, 19].

Qualitative analysis of how the theoretical orientation of an intervention is reflected in participant experiences might contribute to further understanding of how and for whom complex interventions work. It will also illuminate the practical meaning of the theory in this particular context.

## Aim

The aim of the present study was to explore participant experiences of a web-based psycho-educational intervention targeting sexual dysfunction and fertility distress after cancer within the frame of the self-determination theory's concepts of basic psychological needs.

## Methods

### Design

This is a qualitative study using the Framework approach [20] to abductively analyze semi-structured individual interviews. The study is reported according to the COREQ criteria [21] (S1 File).

**The intervention.**    Fertility and sexuality following cancer (Fex-Can) is a Swedish nationwide cohort study with an embedded randomized controlled trial (RCT) [22], testing the effect of a web-based psycho-educational program for either sexual dysfunction or fertility distress. The development and previous feasibility testing of the web-based program have been described in detail elsewhere [23, 24]. Self-determination theory (SDT), as conceived by Ryan and Deci [15] and developed for use within the context of internet interventions for cancer survivors by Pingree et al [18], was used as a theoretical basis for the intervention, which aimed to support participants' competence, relatedness and autonomy in an inclusive and permissive way. The intervention was a 12-week web-based program delivered in six consecutive modules including informative texts and illustrations, automated feedback via quizzes, video vignettes, written quotes from cancer survivors, a moderated discussion forum and exercises inspired by mindfulness and/or cognitive behavioral therapy. The sexuality program also offered a personalized telephone consultation. A sub-sample of participants in the RCT formed the sample for the present study, which was part of the process evaluation of the intervention. Intervention usage, which was also discussed during the interviews, is presented as an extended context description below.

### Sampling

Sampling was purposeful with the intention to represent a diversity of participants in terms of age, gender, cancer diagnoses, relationship status, and level of utilization of the allocated program. Within one month of completing the intervention, 60 participants in the Fex-Can RCT were approached via email and 31 preliminarily accepted to be interviewed. Three withdrew before being interviewed, leaving a final sample of four men and twenty-four women, equally divided between the fertility and sexuality program. For ethical reasons, motives for non-participation were not requested. Some eligible participants spontaneously stated lack of time and not having used the program enough to be able to evaluate it as reasons for declining to be interviewed for the present study.

The proportion of men to women was similar to that in the RCT. The median age was 33 (range 19–40).

## Context

Participants had been diagnosed with either breast, ovarian, cervical or testicular cancer, lymphoma or brain tumors 1–2 years previously. Treatment modalities included surgery, radiotherapy, chemotherapy and long-term hormonal treatment. Although almost all had returned to work or studies, some reported lower capacity, and a few had been on sick leave due to post-treatment depression. Most of the participants were living in heterosexual or same-sex partner relationships. About half had biological children before the cancer diagnosis and one woman was currently expecting. A few women knew their treatment had caused permanent infertility and several more were temporarily infertile or advised against pregnancy due to current hormonal therapy. Sample characteristics are further described in Table 1.

Four different but partly overlapping patterns of approach to the program could be identified, as illustrated below (Fig 1).

Going 'all in' meant being very active, enthusiastic and motivated. Another pattern was about eagerly reporting having processed the texts and completed the modules like a given assignment, out of curiosity or because they wanted to contribute to research. Among the less active, some frankly described that they had not prioritized the program, but not without ambivalence, e.g. regretting not having done more or feeling like having missed out on a time-bound opportunity. Finally, some participants lacked motivation and for varying reasons were not willing or able to take part. The perceived approach was not always consistent with actual use of the program, and activity and motivation also varied during the intervention period, depending both on intervention contents and on external factors. Practical obstacles to using

**Table 1. Participant characteristics (n = 28).** Demographic and clinical variables as reported by participants at the time of the interview.

| Participant characteristics | n |
|---|---:|
| **Age, median (range)** 33 (19–40) | |
| **Gender** | |
| Female | 24 |
| Male | 4 |
| **Diagnosis** | |
| Breast cancer | 10 |
| Gynecological cancer | 7 |
| Lymphoma | 7 |
| CNS tumors | 3 |
| Testicular cancer | 1 |
| **Relationship status** | |
| In a relationship | 21 |
| Single | 7 |
| **Parenthood status** | |
| Had biological children before onset of cancer | 13 |
| Living with partner's biological children | 3 |
| Expecting | 1 |
| Living without children | 11 |
| **Occupational status** | |
| Working full or part time | 22 |
| Studying | 2 |
| Parental leave | 1 |
| Sick leave | 3 |

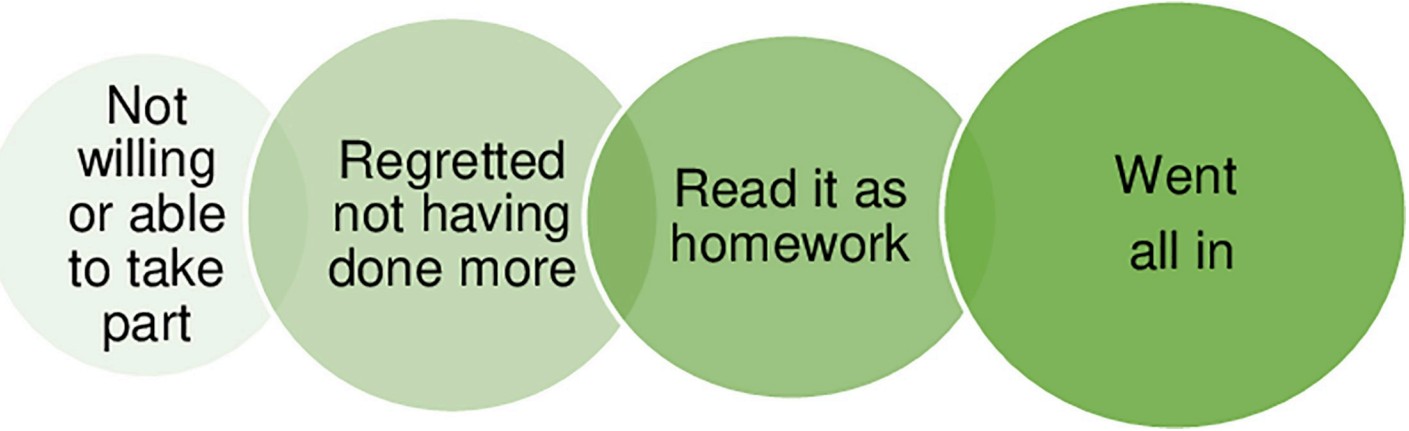

**Fig 1. Typology for using the program.**

the program were mainly related to technical difficulties, perceived lack of time and inconvenient timing of the intervention (Fig 2).

## Procedure

Telephone interviews using a semi-structured interview guide (see S2 File for a summary) were conducted one to two months after the end of the program, between February and July

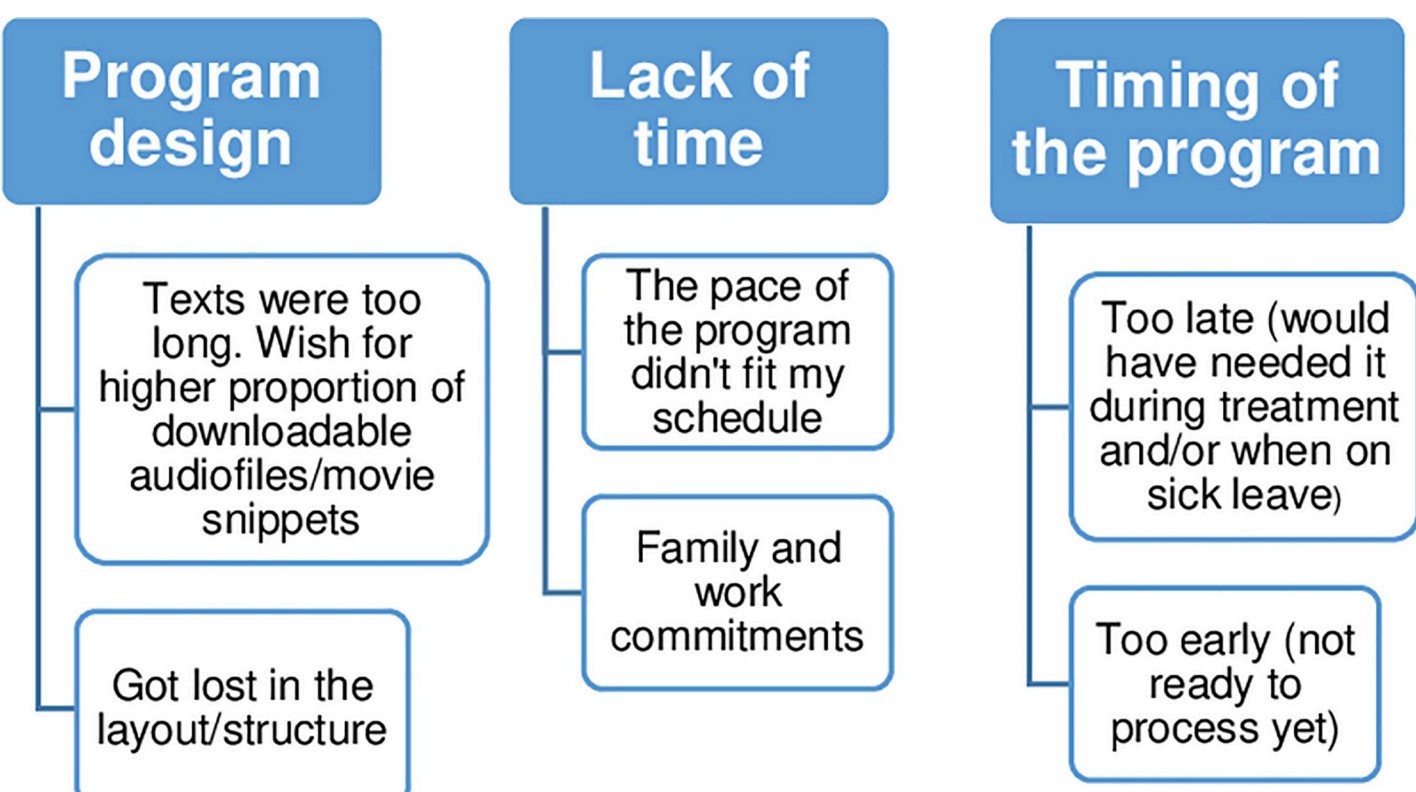

**Fig 2. Reported obstacles to using the web-based psychoeducational program.**

2018. The use of (mobile) telephone interviews as the mode of data collection, enabled participants to freely choose where to be located during the interview. The participants were instructed to place themselves at a location were they would not be disturbed during the interview. The interviews started by the researcher reiterating the purpose of the interview before asking some general questions about demographics and reasons for enrolling in the intervention. Participants were informed about the interviewer's role in the research project at the beginning of the interview, and it was underscored that the purpose of the interview was to understand how the program had been used and evaluate the design of the intervention, not to evaluate participants' performance or activity. More detailed questions followed depending on self-reported usage of the program. Questions were formulated to cover the theoretical concepts without making explicit reference (e.g. "How did you feel reading what others had written in the discussion forum?"). Flexibly rephrasing questions and using probes were among the strategies used to encourage participants to elaborate on issues relevant to them. Respondent validation was ensured by asking participants at the end of the interview if they had anything to add or clarify. Twenty-six out of the 28 interviews were audio-recorded and transcribed verbatim, while two interviews were coded based on the interviewers' detailed notes and/or verbal summaries. Interviews lasted for between 20 and 70 minutes. Interviewers were either registered nurses (CMO, LEE, LW) or licensed psychologists (CL, LL) and had training and/or experience of qualitative interviewing for research or clinical purposes. All co-authors were also involved as researchers in the main intervention study.

## Analysis

The Framework approach [20] involves a circular, iterative process between various levels of abstraction, allowing for abduction, i.e. a combination of deductive and inductive perspectives [25]. NVivo 12 software (QSR International, Melbourne) was used for sorting and displaying data. After familiarization, two interviews were chosen, based on their presumed diversity (age, program, interviewers). These transcripts were inductively/open coded, generating a large number of codes grounded in the data. The first and last author compared and discussed the codes, arriving at a preliminary index. The number of codes was successively reduced to the extent that they fitted within the predefined deductive main themes. This process involved abductive reasoning where inductively generated codes kept being added within the category 'other'. When all the interviews had been coded according to the preliminary index, a final matrix consisting of six main themes and four to six subcategories per theme was visually displayed, discussed, revised and agreed upon in the research team. To summarize and interpret the matrices, the analysis proceeded to thematic charting [20]. Deviant cases needing further analysis were discussed in the research team until agreement was reached on the contents of the three overarching deductive themes and on formulation of the subthemes.

## Ethical considerations

The present study was approved by the Regional Board of Ethics in Stockholm (permit no 2013/1746-31/4). Written informed consent was obtained from all participants prior to enrolment in the RCT, and oral informed consent was recorded during the interview.

## Results

The framework analysis resulted in three themes and three, four and two sub-themes, respectively (Fig 3).

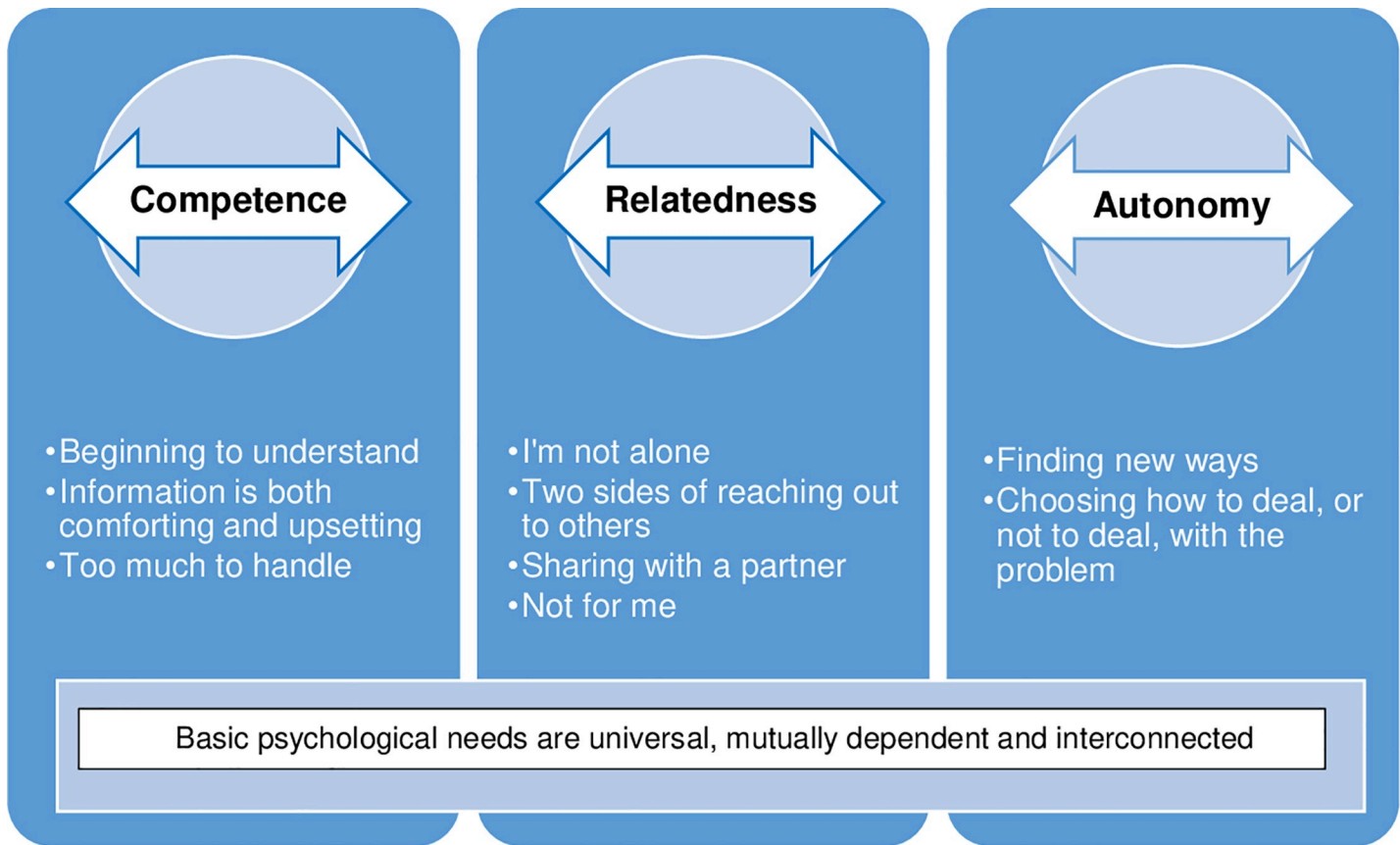

**Fig 3. Overview of themes and subthemes.**

### Competence

Competence was articulated in the way participants reacted to the information included in the program and in their attitude to knowledge concerning cancer, sexual function and fertility.

**Beginning to understand.** The program was appreciated as a check-up opportunity and confirmation of stored-away knowledge, e.g. on anatomy and the hormonal system. The exercises may not have been new to participants, but were recognized as helpful tools. Some were both surprised and relieved to learn that current symptoms and ill-being, e.g. lack of desire, could be related to the cancer treatment. Participants who had felt inferior in relation to healthcare professionals and their disease, now felt more competent and able to formulate new questions. Curiosity about other diagnoses and treatments also contributed to expanding perspectives on one's own situation. The attitude that 'the more information, the better' made participants appreciate the interactive quizzes with immediate feedback. Even the baseline survey used for assessing intervention eligibility had impacted participants in making them reflect about their situation. Not only understanding facts but also understanding one's own reactions facing the cancer trajectory was linked to a new sense of mastery and awareness. The quote 'now I know something about cause and effect', summarized the insight that certain reactions or symptoms were due to the cancer treatment, i.e. legitimate and normal.

**Information is both comforting and upsetting.** A common view was that facts make you calmer, even if the facts per se are negative. The provision of balanced and reliable information was appreciated, even by participants already having a high level of perceived competence.

*Reading cold hard facts. . . maybe it's not what you were expecting. . . the idea that you might not be able to have a child. Even if you already knew it, it hits you hard when you read it there, but then again. . . it is what it is. You shouldn't withhold facts and information either because. . . I think somehow it's good to. . . um, see it in writing too. (10)*

On the other hand, there were also participants who did not feel their competence was strengthened by the program as intended. There was some disappointment about the information provided not being sufficiently detailed and diagnosis-specific, or applicable to their individual situation. Finally, some described hesitating between avoiding all thoughts reminding of their disease, and wanting to learn more. A few even experienced that the program raised new concerns. Strategies used for handling painful and unsolicited information about relapse or infertility sometimes involved quickly shutting down the homepage. Hence, receiving information may have a contradictory effect, on the one hand comforting and calming, on the other hand upsetting.

**Too much to handle.** Participants who were suffering from persistent fatigue and/or cognitive impairment felt impeded in their capacity and motivation for reading and processing large amounts of text. Some described not being able to keep up with the pace of the program or getting lost in the layout. While those who were less motivated to begin with gave up and became inactive, others who 'went all in' tried to find strategies for handling the information. Ways of remembering to log on despite cognitive difficulties included keeping the text message reminders unopened until the timing was right, and setting aside undisturbed time for the program.

## Relatedness

Almost every participant, regardless of program activity, mentioned the positive aspects of sharing experiences with others in similar and sometimes different situations.

**I'm not alone.** The program helped to reduce feelings of loneliness that many participants had been struggling with while encountering difficulties returning to work and managing relationships (professional or intimate) after the treatment period had ended. In this situation of inner feelings of loneliness, frustration and disorientation, the program was like a haven where they could find support and be understood. For some, the affirmation of being 'normal in a new way' was enough, but the discovery that others had similar problems could also be a real eye-opener. One participant even described how, thanks to the program, she had gone from feeling like 'from a different planet' in the eyes of others, to feeling validated as 'normal', and accepted. Another woman regretted not having had access to the program at an earlier point when she was feeling extremely lonely, receiving fertility-threatening treatment for gynecological cancer in the same aftercare ward as new mothers. The program contributed to dissipate loneliness in relation to perceived sexual problems and fertility distress, but people also felt affirmed in their changed outlook on life and their newly gained sense of gratitude about just being alive.

**Two sides of reaching out to others.** Participants described how much it meant to communicate with others, especially through the discussion forum.

*These hearts that you would get, somehow they make you feel supported, that you're not alone in this/. . ./. [the likes and comments in the discussion forum] even made me feel that I would have liked to meet these girls, it reached a level of some kind of affinity. (09)*

The interactivity in the forum, as well as the filmed and written survivor stories in the program, would evoke empathy, sympathy and a sense of connection. There was some hesitation about taking active part in discussions, sometimes linked to feelings of guilt as many perceived

they had 'gotten away too easily' or had been very lucky not getting worse consequences from their disease. Most participants claimed a wish to support others in more difficult situations or contribute to research in a more general way.

**Sharing with a partner.** Partnered participants reflected on the level of involvement of partners, both in the present intervention and in the cancer trajectory in general. Some had let their partner read and use the program whereas others did not dare to do this because it was a research project and partners were not explicitly invited. One male participant who had been substantially inactive in the program described how his female partner following his cancer treatment had been severely depressed and how this had affected their sexual life more than the cancer itself. But since the first chapters in the program were aimed at understanding the impact of cancer on sexuality, he had lost interest in the program and never got to the part linking sexuality after cancer to relationships, and he regretted that. The two women who were in same-sex relationships described their partners as very supportive, contrasting with the heterosexual relationships where support in several cases was conditional. While some of the participants who were living in relationships had very strong bonds with their partners, others had more negative experiences from feeling dismissed when trying to communicate their needs, as if the partner still didn't understand or didn't care. The program had helped leveraging communication about difficult issues and actually improved the relationship to the extent that the partner was allowed in, sympathetic and interested. Still, there was a request for more support regarding partner involvement and relationships.

Participants who were single had somewhat different approaches to the program. One of them had been very active and linked this to the fact that she was not depressed about being single. The other unpartnered participants had been less active in the program and reasoned about partner disclosure in hypothetical ways, i.e. whether a future partner would react negatively when learning about the cancer treatment, but did not perceive they got so much support from the program. However, participants did not describe feeling offended or outright excluded from the bulk of the program for being single.

**Not for me.** There were also instances of feeling overlooked. This could be because the perceived problems were not that distressing or because participants perceived themselves as somehow different or not belonging to the target group. Especially those women who had confirmed infertility felt excluded from many of the themes and discussions concerning fertility distress. There was also the view that the program was interesting in a general way but couldn't relate to the specific topics or problem areas, i.e. felt the program was not 'for them' but for those e.g. younger, living in couples and with at least some fertility potential.

## Autonomy

Autonomy could be seen as moving along a continuum between action and acceptance of changes to sexual function and fertility.

**Finding new ways.** Participants with sexual dysfunction often found new ways to overcome their problems and described how the program had contributed to them taking charge of the situation. One participant said the cancer experience had changed her body image in a more positive direction, and that the exercises in the program had confirmed and accentuated this positive feeling by reducing stereotyped ideals concerning sexuality. Another person remembered how sex would be associated with pain and reluctance, but that the exercises in the program had helped her and her partner find new ways, involving improved communication and respect.

*If I used to feel some desire, I might push the feeling away because "it would only hurt anyway". But then, trying to work with positive thoughts and, well, finding alternative solutions.*

*And also this. . . self-image thing. [Pause] Re-accepting oneself. This is what my body looks like. (13)*

The route towards acceptance was depicted as a winding road of contradictory feelings oscillating between frustration and confidence. The mindfulness exercises as well as informative texts helped in moving closer to acceptance. The program had facilitated the process of coming to terms with issues previously undealt with, and to accept the new conditions.

**Choosing how to deal or not to deal with the problem.**   One aspect of autonomy was reflected in the conscious choice to take action in a certain direction, or making the decision *not* to act. Several participants described sexual and reproductive issues in terms of work that needed to be done bit by bit, or a responsibility to 'do whatever I can to improve the situation', 'know where to turn to' and to 'work actively'. Responsibility for one's own situation was described in both positive and negative terms, as a way of freely moving in a valued direction or as a heavy yoke carried by no one else.

An important reason for ambivalence towards the program was feeling reluctant to even think about the diagnosis and its consequences, such as confirmed infertility. Some expressed it as a more or less conscious choice to postpone dealing with certain issues until 'later', waiting to get stronger or for things to go back to an imagined 'normal' again. Inclusion in the intervention sometimes strengthened autonomy in unexpected ways. One woman, having hardly used the program because 'it was so instinctively painful', described how the intervention still had made her start processing emotions related to possible infertility, and finally realize that she needed individual psychotherapy.

## Discussion

The aim of this study was to describe participant experiences of a web-based program for alleviating sexual dysfunction and fertility distress after cancer and to explore these within the frame of the self-determination theory concepts of basic psychological needs. The main themes *competence*, *relatedness* and *autonomy* were reflected in nuanced, in-depth yet often contradictory descriptions of participant experiences.

*Competence* was linked to the amount and content of information delivered in the program, in relation to personal expectations and needs. While participants generally appreciated having access to reliable information, certain facts would evoke anxiety, leading to avoidance strategies. Conversely, some described how the program had made them search more actively and consciously for what they needed, implying increased *autonomy*. The subtheme 'information overload' may be reflective of the program conceivers' underestimation of cognitive difficulties encountered in cancer survivorship, increasingly recognized in research [26–28]. Some participants found strategies for handling the flow of information, and thereby could make reflective choices on how to take part in the program and how to handle their situation. The intertwining of the concepts of autonomy and competence here appeared clearly. However, the principle of respecting autonomous choices [17] could conflict with intervention hypotheses and end-outcomes expecting change and improvement. Therefore, if the intervention was successful in nurturing autonomy, it is a logical result that some participants decided *not* to act to improve their situation, no matter helpful that would have been in the long run.

Internet use among young adults with cancer has previously been found to be a complex phenomenon mainly driven by negative emotions [29]. The findings of the present study challenge this view, by depicting a *relatedness*-supportive environment where participants could feel at ease, contributing to a deep sense of solidarity and of belonging to an understanding and benevolent community. The program thus dissipated part of the loneliness which has

been reported as a major concern for young patients with cancer experiencing fertility distress [30].

Despite efforts to achieve an inclusive intervention, the program was not perceived by all participants as supportive, suggesting some mismatch between participants' perceived problem areas, expectations and level of tailoring of the program. This is in keeping with the literature where several studies [29, 31] emphasize that patients suffering from long-term sequelae of treatment request internet-delivered detailed, personalized information and support. Our findings may imply that participants who chose a more reluctant approach to the program possibly did not feel their individual information and support needs were met. One difficulty in analyzing this phenomenon was that evaluative aspects of utilization were difficult to separate from emotional experiences. It is known that motivation and curiosity contribute to the use of web-based behavior change interventions for the general population [32] and also that experiences of internet interventions for cancer survivors may be negatively influenced by time since diagnosis, as well as lack of time and perceived workload [33]. Previous research also points out technical problems, personal expectations and varying needs as possible reasons for non-use [34]. All of these aspects were confirmed in our findings, i.e. participants brought up that the program did not always fit with their current needs, and some were also overwhelmed by the amount of information and the pace of the program. Also, despite prior feasibility testing [24], some technical difficulties persisted and participants expressed that the design of the website was too complicated. In general, such problems were easily solved in cases where participants had chosen to contact the support function.

In the present study, no clear patterns were seen between these described barriers to utilization, and the perceived or actual adherence to the program. It had been hypothesized that the level of adherence would be related to severity of problems at baseline, mediated by motivation for change, and that there would be a dose-response effect. Our findings suggest that there may be other factors influencing the way an intervention is utilized. Participants who described themselves as inactive could sometimes, just like those who went 'all in', give rich accounts of how their perceptions of fertility and sexuality issues had evolved during the intervention. Others described how their reflections were triggered by just receiving the baseline survey. The intervention seems to have been a catalyst for a movement that was often already there.

## Methodological considerations

The main advantage of the Framework approach in this study was the possibility to process large amounts of qualitative data in a transparent and structured way [20]. Our study thereby complements the results from a nationwide RCT with qualitative data from a maximum diversity sample enabling representational generalization to the population of the main study [20]. Validity-enhancing measures and active use of the researchers' preunderstanding included the members of the research team reading and commenting on each other's transcripts during the iterative analysis process, and discussing the results and interpretations with researchers external to the project.

**Study limitation.** The double role of the interviewers as conceivers and evaluators of the program may have prevented participants from displaying negative attitudes, despite encouragement to the contrary. Low-users rarely spoke specifically about the program, and tended to talk more generally about the cancer trajectory or about research. In the cases where non-use seemed to be associated with bad conscience and guilt it was even more difficult to discuss experiences in-depth. The semi-structured approach may also have been suboptimal for uncovering all potentially relevant issues with both high- and low-users. To enhance reliability

and validity in this respect, active probing was used and at the end of each interview, participants were asked if they wanted to add or clarify anything.

Also, very few men, all low-users, accepted to be interviewed. This is not surprising since men and especially younger men generally tend to have lower participation rates in survey-based research, a phenomenon that has also been confirmed in young adult cancer survivors [35]. The ratio of men to women was already low in the main study and had decreased even further in the present subsample. While it is known that men do suffer from fertility distress and sexual dysfunction after cancer [6, 36], previous results from our research group suggest that young men with testicular cancer may not be as severely affected by sexual dysfunction and fertility distress as women with cancer, as men with other malignant disorders or as hypothesized on the basis of data from earlier studies [37]. This conclusion is supported by our findings, but does not fully explain why the men who were interviewed for the present study (one of which had testicular cancer, the other three hematologic cancers), despite scoring above the cut-off level for the intervention, stated they had no particular sexual or fertility problems and therefore substantially felt the program was not "for them". Another potential partial explanation for the low participation rate of men in the present study is that male cancer patients might prefer "male only" and diagnosis-specific support groups [38]. The lack of male perspective nevertheless limits inferential generalization [20] to the population of male cancer survivors, and warrants further research focusing on the specific concerns of young adult men with diagnoses such as brain tumors or hematologic malignancies, where available treatment options likely are detrimental to sexual and reproductive function [2].

Finally, while a deductive approach is not recommended in the original Framework approach [20], it has been applied in various studies [33, 39–41] and Framework clearly supports simultaneous sorting and interpretation of data. A deductive-abductive methodology involves the risk of over-interpreting data according to the predefined abstract themes. To counteract the risk of theoretical overreliance, the authors had frequent discussions concerning the interpretation of findings, including checking back against raw data as recommended [20].

## Conclusions

The Fex-Can intervention was generally perceived as supportive of the needs for competence and relatedness. Descriptions of autonomy were more ambiguous. While all three basic psychological needs are mutually dependent [15], a preliminary hypothesis generated from the present study is that satisfaction of relatedness seems to be of special importance to young cancer survivors. Nevertheless, the diversity and contradiction in descriptions of basic need satisfaction is also a main conclusion from the present study. Further research is needed to qualitatively and quantitatively determine the relationship between individual psychological and treatment-related characteristics, satisfaction of basic psychological needs and responses to complex interventions.

## Clinical implications

The results of the present study indicate that healthcare professionals need to consider individual levels of competence (knowledge and cognitive capacity) as well as the need for relatedness when discussing sexuality and fertility in clinical care of young adults with cancer. Personalized information when requested, as well as arenas for peer and/or relational support, may enhance people's capacity for autonomous decisions regarding sexual and reproductive health. There should exist technical solutions to develop tailored support delivered via the healthcare system or independently available to patients in the survivorship phase. Our findings may help in optimizing such theory-based interventions to suit the needs of young cancer survivors.

## Supporting information

**S1 File.**
(DOCX)

**S2 File.**
(DOCX)

## Acknowledgments

The authors wish to thank all participants of the Fex-Can program who so generously shared their experiences when interviewed for the purpose of this study.

## Author Contributions

**Conceptualization:** Claire Micaux Obol, Claudia Lampic, Lena Wettergren, Lisa Ljungman, Lars E. Eriksson.

**Data curation:** Claire Micaux Obol.

**Formal analysis:** Claire Micaux Obol, Lisa Ljungman, Lars E. Eriksson.

**Funding acquisition:** Claudia Lampic, Lena Wettergren.

**Investigation:** Claire Micaux Obol, Claudia Lampic, Lena Wettergren, Lisa Ljungman, Lars E. Eriksson.

**Methodology:** Lisa Ljungman, Lars E. Eriksson.

**Project administration:** Claire Micaux Obol.

**Supervision:** Lars E. Eriksson.

**Validation:** Claudia Lampic, Lena Wettergren, Lisa Ljungman, Lars E. Eriksson.

**Writing – original draft:** Claire Micaux Obol.

**Writing – review & editing:** Claire Micaux Obol, Claudia Lampic, Lena Wettergren, Lisa Ljungman, Lars E. Eriksson.

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
