## [Decision Letter · Decision Letter 0]

12 May 2020

PONE-D-20-08318

Experiences of a web-based psycho-educational intervention targeting sexual dysfunction and fertility distress in young adults with cancer - a self-determination theory perspective.

PLOS ONE

Dear Mrs Micaux Obol,

Thank you for submitting your manuscript to PLOS ONE. After careful consideration, we feel that it has merit but does not fully meet PLOS ONE’s publication criteria as it currently stands. Therefore, we invite you to submit a revised version of the manuscript that addresses the points raised during the review process.

We would appreciate receiving your revised manuscript by Jun 26 2020 11:59PM. To enhance the reproducibility of your results, we recommend that if applicable you deposit your laboratory protocols in protocols.io, where a protocol can be assigned its own identifier (DOI) such that it can be cited independently in the future. For instructions see: http://journals.plos.org/plosone/s/submission-guidelines#loc-laboratory-protocols

We look forward to receiving your revised manuscript.

Kind regards,

Elena Ambrosino

Academic Editor

PLOS ONE

Journal Requirements:

Reviewers' comments:

Reviewer's Responses to Questions

**Comments to the Author**

1. Is the manuscript technically sound, and do the data support the conclusions?

Reviewer #1: Yes

Reviewer #2: Yes

Reviewer #3: Yes

Reviewer #4: Yes

2. Has the statistical analysis been performed appropriately and rigorously? 

Reviewer #1: Yes

Reviewer #2: I Don't Know

Reviewer #3: N/A

Reviewer #4: N/A

3. Have the authors made all data underlying the findings in their manuscript fully available?

Reviewer #1: No

Reviewer #2: Yes

Reviewer #3: No

Reviewer #4: Yes

4. Is the manuscript presented in an intelligible fashion and written in standard English?

Reviewer #1: Yes

Reviewer #2: Yes

Reviewer #3: Yes

Reviewer #4: Yes

5. Review Comments to the Author

Reviewer #1: Additional comments are as follows:

Q1 How did you measure the level of psychological situation like; loneliness and normalizing problems your patients? And when should be the specialist stop the program?

Q2 There are physiological different between women and men for sexual health after treatment, I think it is very important add some sentences about effect of treatment on sexual health in men and woman. And How treatment (Radiation, chemotherapy Therapy) Can Affect the Sex Life?

Q4 Line 103 P5: authors state that "Practical obstacles to using the program were mainly related to technical difficulties ..." do you have suggestions for solution these problems?

Q3 Line 173 P8: authors state that "On the other hand, some participants did not feel their competence …" how many patients?

Note: As commented above, overall the study is nice and has big potential.

Reviewer #2: The subject is suitable for the growing use of internet for medical and healthcare, especially considering the current crisis internationally.

Any additional research info finding support for cancer are much needed.

Reviewer #3: Overall is an interesting study, but I have the following concerns and comments:

- Introduction is clear and well defined as well as the objectives.

- The methods section could be improved with the adoption of a research protocol. In this case the COREQ is indicated, despite the fact that the manuscript contains almost every aspect covered by the protocol. Even though, such adoption is recommended for compliance purposes.

- The data presented in the “Context” section could be improved with further data about the participants, especially with the specific information instead of “some reported”, or “About half”, or “A few”. Such precision is relevant since the discussion also relies on these aspects. In other words, the authors mention characteristics they want and expect about the subjects, but they do not exactly present the characteristics of those who participated.

- Discussion is broad and goes to the point.

- References are relevant and up to date.

Reviewer #4: Overall, the study presents good points about cancer survivors’ sexual and reproductive health.

Although the self-determination theory perspective and deductive approach, according to three predetermined themes, can lead to the loss of some important data and concerns of target population.

However, despite the authors substantial explanations in study limitations, there is still a big question about gender proportion in the main RCT.

6. PLOS authors have the option to publish the peer review history of their article (what does this mean?). If published, this will include your full peer review and any attached files.

Reviewer #1: Yes: Mohamed Hadi Mohamed Abdelhamid

Reviewer #2: Yes: Dr Lily Abedipour MD

Reviewer #3: No

Reviewer #4: Yes: Seyed Ali Azin

---

## [Author Response · Author response to Decision Letter 0]

16 Jun 2020

Review Comments to the Author

We wish to thank the reviewers for very valuable comments. Please see or answers in italics directly after each comment. 

Reviewer #1: Additional comments are as follows:

Q1 How did you measure the level of psychological situation like; loneliness and normalizing problems your patients? And when should be the specialist stop the program?

 Answer Q1. The present study is part of a larger RCT with measurements at baseline, post-intervention (directly after end of program) and short-term follow up (12 weeks after end of program). Participants (n=265) were assessed with PROMs for physical and mental health as part of the quantitative evaluation of the intervention. These results will be published separately. Concerning the subsample of 28 men and women in the present qualitative study, the individual levels on these outcome measures were not taken into consideration and therefore we have not sought or reported the exact scores of the 28 participants. In a few of the interviews, negative feelings such as anxiety, grief, loneliness were expressed. In these cases, the interviewer(s) took care to separate the research interview from a more informal conversation at the end of the interview, intending to provide some support and referral to counselling or other healthcare professionals if needed. 

Q2 There are physiological different between women and men for sexual health after treatment, I think it is very important add some sentences about effect of treatment on sexual health in men and woman. And How treatment (Radiation, chemotherapy Therapy) Can Affect the Sex Life?

 Answer Q2. We thank the reviewer for this valuable comment and have extended the Background section with information on the medical and physiological consequences of cancer and its treatment on the sexual health of both men and women,

Q4 Line 103 P5: authors state that "Practical obstacles to using the program were mainly related to technical difficulties ..." do you have suggestions for solution these problems?

 Answer Q4. We have complemented the context description with a figure illustrating stated reasons for non-use of the program, including technical problems. We have also extended the discussion section with a paragraph on the implications of technical difficulties. 

Q3. Line 173 P8: authors state that "On the other hand, some participants did not feel their competence …" how many patients?

 Answer Q3. Since the present study has a qualitative design, we want to be cautious about making quantitative claims when describing the results. Instead we strive for describing the whole spectrum of participants’ experiences. In order to clarify this passage, we have rephrased some sentences but without adding quantitative descriptors. 

Note: As commented above, overall the study is nice and has big potential.

Reviewer #3: Overall is an interesting study, but I have the following concerns and comments:

- Introduction is clear and well defined as well as the objectives.

- The methods section could be improved with the adoption of a research protocol. In this case the COREQ is indicated, despite the fact that the manuscript contains almost every aspect covered by the protocol. Even though, such adoption is recommended for compliance purposes.

We thank the reviewer for this suggestion and have added the COREQ checklist to the supplementary file. The manuscript and supplementary files have been revised with addition of the few points from the COREQ that were not covered in the previous version. 

- The data presented in the “Context” section could be improved with further data about the participants, especially with the specific information instead of “some reported”, or “About half”, or “A few”. Such precision is relevant since the discussion also relies on these aspects. In other words, the authors mention characteristics they want and expect about the subjects, but they do not exactly present the characteristics of those who participated.

 Answer: We agree that the Context section was too brief and have incorporated a table detailing as much participant data as possible without compromising the confidentiality and privacy of participants. 

Reviewer #4: Overall, the study presents good points about cancer survivors’ sexual and reproductive health.

Although the self-determination theory perspective and deductive approach, according to three predetermined themes, can lead to the loss of some important data and concerns of target population.

However, despite the authors substantial explanations in study limitations, there is still a big question about gender proportion in the main RCT.

 Answer: We thank the reviewer for this very relevant comment. The question of gender proportion is also our concern and we have added a few more sentences in the Discussion section about this issue Certainly, it will also be discussed in our future publications reporting the results and process of the RCT.

---

## [Decision Letter · Decision Letter 1]

1 Jul 2020

Experiences of a web-based psycho-educational intervention targeting sexual dysfunction and fertility distress in young adults with cancer - a self-determination theory perspective.

PONE-D-20-08318R1

Dear Dr. Micaux Obol,

We’re pleased to inform you that your manuscript has been judged scientifically suitable for publication and will be formally accepted for publication once it meets all outstanding technical requirements.

Kind regards,

Elena Ambrosino

Academic Editor

PLOS ONE

Additional Editor Comments (optional):

Reviewers' comments:

Reviewer's Responses to Questions

**Comments to the Author**

1. If the authors have adequately addressed your comments raised in a previous round of review and you feel that this manuscript is now acceptable for publication, you may indicate that here to bypass the “Comments to the Author” section, enter your conflict of interest statement in the “Confidential to Editor” section, and submit your "Accept" recommendation.

Reviewer #1: All comments have been addressed

Reviewer #3: All comments have been addressed

2. Is the manuscript technically sound, and do the data support the conclusions?

Reviewer #1: Yes

Reviewer #3: Yes

3. Has the statistical analysis been performed appropriately and rigorously? 

Reviewer #1: Yes

Reviewer #3: I Don't Know

4. Have the authors made all data underlying the findings in their manuscript fully available?

Reviewer #1: Yes

Reviewer #3: Yes

5. Is the manuscript presented in an intelligible fashion and written in standard English?

Reviewer #1: Yes

Reviewer #3: Yes

6. Review Comments to the Author

Reviewer #1: No comment, the authors have adequately addressed my comments raised in a previous round of review and I feel that this manuscript is now acceptable for publication.

Reviewer #3: (No Response)

7. PLOS authors have the option to publish the peer review history of their article (what does this mean?). If published, this will include your full peer review and any attached files.

Reviewer #1: **Yes: **Mohamed Hadi Mohamed Abdelhamid

Reviewer #3: **Yes: **Bianca Bianco

---

## [Editor Report · Acceptance letter]

7 Jul 2020

PONE-D-20-08318R1 

Experiences of a web-based psycho-educational intervention targeting sexual dysfunction and fertility distress in young adults with cancer - a self-determination theory perspective 

Dear Dr. Micaux Obol:

I'm pleased to inform you that your manuscript has been deemed suitable for publication in PLOS ONE. Congratulations! Your manuscript is now with our production department. 

Kind regards, 

on behalf of

Dr. Elena Ambrosino 

Academic Editor

PLOS ONE